# Synthesis of *N*-acyl sulfenamides via copper catalysis and their use as *S*-sulfenylating reagents of thiols

Ziqian Bai [1], Shiyang Zhu[1], Yiyao Hu[1], Peng Yang[1], Xin Chu[1], Gang He [1], Hao Wang [1] & Gong Chen [1,2,3]

Sulfur–heteroatom bonds such as S−S and S−N are found in a variety of natural products and often play important roles in biological processes. Despite their widespread applications, the synthesis of sulfenamides, which feature S−N bonds that may be cleaved under mild conditions, remains underdeveloped. Here, we report a method for synthesis of *N*-acyl sulfenamides via copper-catalyzed nitrene-mediated *S*-amidation reaction of thiols with dioxazolones. This method is efficient, convenient, and broadly applicable. Moreover, the resulting *N*-acetyl sulfenamides are highly effective *S*-sulfenylation reagents for the synthesis of unsymmetrical disulfides under mild conditions. The *S*-sulfenylation protocol enables facile access to sterically demanding disulfides that are difficult to synthesize by other means.

Thiol groups play an indispensable role in both natural and synthetic molecules[1–5]. Their facile coupling with carbon groups via strong S−C bonds has enabled a myriad of methods for the synthesis and modification of molecules of varied size and complexity[6,7]. Besides S−C bonds, thiols can also form weaker S−heteroatom bonds such as S−S bond in disulfides and S−N bond in sulfenamides−both of these have unique redox properties and can be selectively cleaved under mild conditions[8–10].

Disulfide moieties are commonly found in synthetic compounds and natural products and have been widely adopted in the construction of biopharmaceuticals such as antibody-drug conjugates (ADC) (Fig. 1a)[11–17]. In recent decades, the chemistry of disulfide synthesis via either direct S−S coupling or reactions with masked S−S precursors has been greatly advanced[18–30]. Nevertheless, challenges remain for the synthesis of unsymmetrical disulfides−especially those with significant steric hindrance around the α carbons[31–34]. Such bulky substitution can greatly enrich the stereochemical features of disulfide moieties and influence their reactivity. For example, hindered alkyl S−S linkers are used in the ADC Myloarg and mAb-DM1 to achieve better therapeutic properties[15–17].

In comparison with S−S bonds of disulfides, S−N bonds of sulfenamides are more polarized with the S atom being more electrophilic[35–38]. The reactivity of the S−N bonds in sulfenamides can be fine-tuned through substitutions on either N or S atoms. As outlined in Fig. 1b, sulfenamides have many useful applications such as vulcanization additives in rubber industry, intermediates in synthesis of various sulfur-containing compounds, and prodrugs in medicinal chemistry[36]. They are also found in natural products such as scorodophlone A and play active roles in living systems such as post-translational modifications of proteins (PTM) during oxidative stress response[37]. Existing methods for sulfenamide synthesis mostly rely on the nucleophilic substitution of sulfenyl compounds bearing a suitable leaving group (LG), e.g., sulfenyl chloride with amines[39–50]. More recently, copper-catalyzed S−N coupling of thiol or disulfides and amines offer a more straightforward synthesis[45,48]. However, these coupling methods are mostly limited to the reactions of aryl thiols. The syntheses of *S*-alkyl sulfenamides are often plagued by the side reactions of alkyl thiols such as S−S homo coupling or limited substrate scope. Practical methods for synthesis of *S*-alkyl sulfenamides are thus greatly needed to broaden the application of sulfenamides. Herein, we report a method for the synthesis of *N*-acyl sulfenamides via copper-catalyzed nitrene-mediated *S*-amidation of both alkyl and aryl thiols with various carboxylic acid-derived 1,4,2-dioxazol-5-ones. Moreover, the resulting *N*-acetyl sulfenamides provide a class of powerful

[1]State Key Laboratory and Institute of Elemento-Organic Chemistry, College of Chemistry, Nankai University, Tianjin 300071, China. [2]Frontiers Science Center for New Organic Matter, Nankai University, Tianjin 300192, China. [3]Haihe Laboratory of Sustainable Chemical Transformations, Tianjin 300192, China. ✉e-mail: hao@nankai.edu.cn; gongchen@nankai.edu.cn

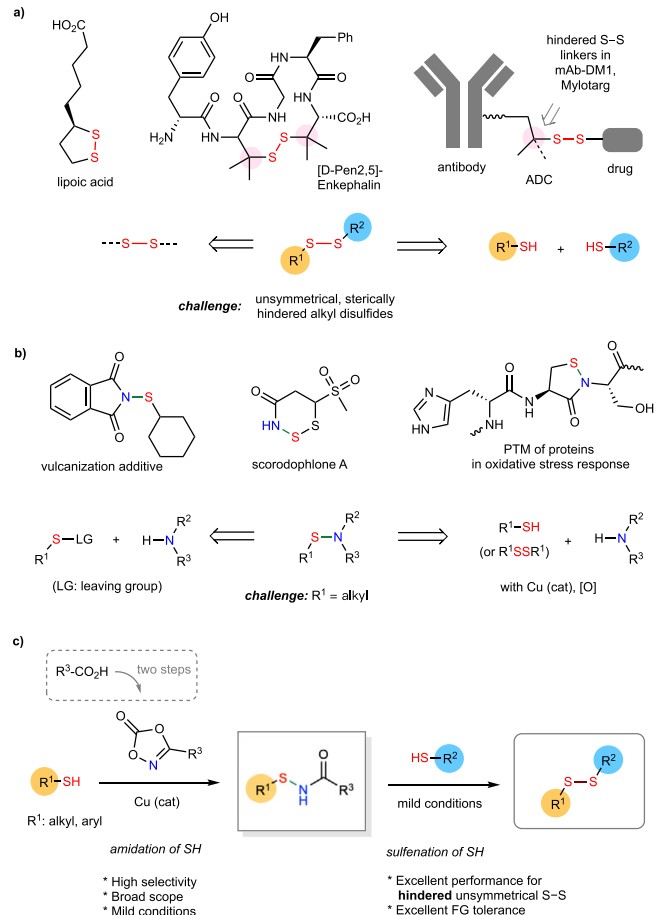

**Fig. 1 | Occurrence and construction of weak S–S and S–N bonds in disulfides and sulfenamides. a** Occurrence and common synthesis strategy of disulfides. **b** Occurrence and common synthesis strategy of sulfenamides. **c** Cu-catalyzed nitrene-mediated amidation of thiols and the use of *N*-acetyl sulfenamide in making unsymmetrical disulfides (this work). ADC antibody-drug conjugates, PTM post-translational modifications.

*S*-sulfenylating reagents for the synthesis of challenging unsymmetrical disulfides under mild conditions (Fig. 1c).

## Results and discussion

### Background on the nitrene-mediated *S*-imidation reactions

Nitrene-mediated *S*-imidation reactions of sulfides and sulfoxides with various nitrene precursors such as *N*-haloamides, oxaziridines, 1,4,2-dioxazol-5-ones, azides, and iminoiodinane derivatives under both metal-free and metal-catalyzed conditions have been well studied[51–61]. A variety of metal catalysts including iron, copper, rhodium, ruthenium, silver, and manganese-based complexes have been successfully employed in these transformations[54–56]. Among the nitrene precursors, dioxazolones have attracted considerable attention in recent years[62–64]. These reagents can be readily prepared from the corresponding alkyl and aryl carboxylic acids via a two-step sequence of coupling with hydroxyamine and cyclization. In the 1960s, Sauer and Mayer reported the first prototype *S*-imidation reaction of dioxazolones with excess dimethyl sulfoxide as a solvent under thermal or photochemical conditions[51]. As a critical new advance, Bolm showed that the *S*-imidation reaction of sulfides and sulfoxides with dioxazolones can proceed efficiently under mild light-induced Ru-catalyzed conditions at room temperature (rt)[57]. More recently, Uchida reported the enantioselective imidation of sulfides with dioxazolones under Ru catalysis[60]. In sharp contrast to the sulfide and sulfoxide substrates, the analogous *S*-amidation or imidation reactions of thiols are very rare[65].

### Cu-catalyzed *S*-amidation of thiols with dioxazolones

Besides the *S*-amidation of sulfur, the utility of dioxazolones as acyl nitrene precursors have also been demonstrated in other important transformations such as C–H amidation reactions in recent years[66–73]. We previously showed that dioxazolones can enable efficient amidation of arylamines and various phosphorus compounds to construct N–N and P–N bonds under the catalysis of iridium or iron[74,75]. Motivated by the prevalence of thiol group in both small molecules and peptides, we questioned whether dioxazolones can react with SH to give the sulfenamide product under proper metal-catalyzed conditions. We thus first studied the reaction of model substrates isopropylthiol **1** and 3-methyldioxazolone **2** (Table 1). We were pleased to find that several kinds of metal complexes including Ir, Fe, Cu, Ru, Co, and Ni can give the desired *S*-amidation products in varied yield and selectivity (see Supplementary Table 1 for details). For example, the reaction of **1** (2.0 equiv), **2** (1.0 equiv), and 2.5 mol% of [RuCp*Cl₂]₂ in 1,2-dichloroethane (DCE) at rt for 24 h gave 16% of the desired *N*-acetyl sulfenamide product **3a** along with 39% of the di-amination product **3b** and 15% of disulfide **3c** (entry 1). Control experiments showed that **3a** can undergo further amidation with **2** to give **3b** under the same reaction conditions. Compound **3a** can also undergo nucleophilic substitution with **1** to give disulfide **3c** in the absence of Ru catalyst. The use of Fe catalysts such as FeCl₂·4H₂O and 2,6-bis[1-(2,6-diisopropylphenylimino)ethyl] pyridine iron(II) dichloride (FeCl₂·PDI) gave a mixture of **3a**, **3b**, and **3c** (entries 2, 3).

Interestingly, little **3b** was formed when copper catalysts were used (entries 4–9). The reaction of **1** and **2** with 5 mol% CuOAc in *n*-hexane gave **3a** in 83% yield (entry 6). However, the performance of CuOAc-catalyzed reaction dropped considerably in polar solvents such as CH₃CN and hexafluoroisopropanol (HFIP), which are important for dissolving polar substrates (entries 7, 8; see Supplementary Table 2 for more details). Gratifyingly, the reactivity in polar solvents can be restored by using copper catalysts bearing *N*-heterocyclic carbene (NHC) ligands. The reaction of **1** (2.0 equiv) and **2** (1.0 equiv) with 5 mol% of [1,3-bis(2,6-diisopropylphenyl)imidazolylidene] copper(I) chloride (IPrCuCl) at rt gave **3a** in 91% isolated yield along with trace amount of **3c** (entry 9). Interestingly, the addition of water as co-solvent did not significantly affect the reaction (entry 13). Reaction with IPrCuCl in *n*-hexane or DCE solvent gave lower yield (entries 11 and 12). Modification of the NHC ligand or replacing the halide ion of IPrCuCl did not give markedly improved results (entries 14–18). Lowering the amount of **1**, reaction temperature, or concentration of reactants all led to diminished yield of **3a** (entries 10, 19, 20).

Primary thiols have high nucleophilicity and could more easily react with sulfenamides to form a disulfide side product than secondary and tertiary thiols. Indeed, the *S*-amidation reaction of model primary thiol Boc-protected cysteine methyl ester **4** (2.0 equiv) with **2** (1.0 equiv) under optimized conditions mentioned above gave a mixture of the desired sulfenamide **5a** (31%) and a significant amount of disulfide **5c** (69%) (Table 2, entry 1). No di-amination product **5b** was detected. Reversing the ratio of **4/2** to 1:2 increased the yield of **5a** to 57% (entry 2). Interestingly, the addition of a small amount of Ag₂CO₃ additive (10 mol%) further improved the yield of **5a** to 74% yield (entry 7). The reaction was performed at the gram scale (15 mmol) and gave **5a** in 72% isolated yield. The Cα chiral integrity of cysteine in **5a** was unaffected (>99.9% ee). Other Ag additives, Cu catalysts, or increased amount of Ag₂CO₃ did not lead to markedly improved results (entries 3–5 and 8; see Supplementary Table 4 for more details). The role of Ag₂CO₃ additive is unclear at the moment. We suspect that Ag₂CO₃ may slightly lower the concentration of free thiol reactant by reversible complexation, thus reducing the formation of undesired disulfide side product[76,77].

This Cu-catalyzed *S*-amidation reaction of thiol with dioxazolone likely follows a similar mechanism to metal-catalyzed nitrene-

## Table 1 | Metal-catalyzed *S*-amidation of secondary thiol 1 with 2

| Entry | Conditions | Yield (%)[a] 3a/3b/3c |
|---|---|---|
| 1 | **1** (2 equiv), **2** (1 equiv), [Cp*RuCl₂]₂ (2.5 mol%), DCE | 16/39/15 |
| 2 | **1** (2 equiv), **2** (1 equiv), FeCl₂·4H₂O (5 mol%), DCE | 21/51/9 |
| 3 | **1** (2 equiv), **2** (1 equiv), FeCl₂·PDI (5 mol%), DCE | 17/32/5 |
| 4 | **1** (2 equiv), **2** (1 equiv), Cu(OAc)₂ (5 mol%), DCE | 27/ND/4 |
| 5 | **1** (2 equiv), **2** (1 equiv), CuOAc (5 mol%), DCE | 53/ND/8 |
| 6 | **1** (2 equiv), **2** (1 equiv), CuOAc (5 mol%), *n*-hexane | 83/ND/11 |
| 7 | **1** (2 equiv), **2** (1 equiv), CuOAc (5 mol%), CH₃CN | 29/ND/9 |
| 8 | **1** (2 equiv), **2** (1 equiv), CuOAc (5 mol%), HFIP | 11/ND/4 |
| 9 | **1** (2 equiv), **2** (1 equiv), IPrCuCl (5 mol%), HFIP | 95 (91[b])/ND/5 |
| 10 | **1** (1 equiv), **2** (1 equiv), IPrCuCl (5 mol%), HFIP | 67/ND/1 |
| 11 | **1** (2 equiv), **2** (1 equiv), IPrCuCl (5 mol%), *n*-hexane | 37/ND/9 |
| 12 | **1** (2 equiv), **2** (1 equiv), IPrCuCl (5 mol%), DCE | 57/11/19 |
| 13 | **1** (2 equiv), **2** (1 equiv), IPrCuCl (5 mol%), HFIP/H₂O (10/1) | 86/ND/3 |
| 14 | **1** (2 equiv), **2** (1 equiv), SIPrCuCl (5 mol%), HFIP | 86/ND/3 |
| 15 | **1** (2 equiv), **2** (1 equiv), IMesCuCl (5 mol%), HFIP | 32/ND/ND |
| 16 | **1** (2 equiv), **2** (1 equiv), IPrCuBr (5 mol%), HFIP | 93/ND/2 |
| 17 | **1** (2 equiv), **2** (1 equiv), IPrCuI (5 mol%), HFIP | 68/ND/2 |
| 18 | **1** (2 equiv), **2** (1 equiv), IAdCuCl (5 mol%), HFIP | 85/ND/2 |
| 19 | **1** (2 equiv), **2** (1 equiv), IPrCuCl (5 mol%), HFIP, 10 °C | 8/ND/ND |
| 20 | **1** (2 equiv), **2** (1 equiv), IPrCuCl (5 mol%), HFIP (0.05 M) | 67/ND/3 |

Catalysts: FeCl₂·PDI; IPrCuCl (X = Cl), IPrCuBr (X = Br), IPrCuI (X = I); SIPrCuCl; IMesCuCl (R = Mes), IAdCuCl (R = Ad)

*ND* not detected, *dipp* 2,6-diisopropylphenyl, *Mes* mesityl, *Ad* admantyl.
[a]All screening reactions were carried at a 0.2 mmol scale; yields were based on ¹H NMR analysis of crude reaction mixture after workup.
[b]Isolated yield at a 0.4 mmol scale.

mediated *S*-imidation of sulfides (Fig. 2). Decarboxylation of dioxazolone at the Cu center of catalyst first forms a Cu-nitrenoid intermediate. Reaction of thiol with Cu-nitrenoid and the subsequent protonolysis furnishes the *N*-acyl sulfenamide. The details of the S−N forming step are still unclear at the moment. We suspect the thiol can directly attack the electrophilic N atom of Cu-nitrenoid. It is also possible that thiol group forms a complex with the Cu center before attacking the N atom[74]. The inherent reactivity of the NHC-bound Cu-nitrenoid intermediate might be critical to preventing further *S*-imidation of the *N*-acyl sulfenamides.

Figure 3 shows that the Cu-catalyzed *S*-amidation reaction exhibits excellent substrate scope for both thiols and dioxazolones under optimized conditions. Reactions of 2° and 3° alkyl thiols generally worked well to give the desired sulfenamide products in good to excellent yield under condition **[A]** in which dioxazolone was used as the limiting reagent. For example, the reaction of

cyclohexanethiol with **2** gave **6** in excellent yield. The reaction of the 2° thiol derived from β-D-glucose tetraacetate with **2** gave **7** whose structure was confirmed by X-ray crystallographic analysis. The reaction of menthol-derived 3° thiol with **2** gave **15** in 62% yield. Aryl and primary 1° alkyl thiols were prone to form more disulfide side products and needed to be used as the limiting reagents (condition **[B]**). A variety of amide groups can be added to the sulfur atom of cysteine in moderate to good yield. Notably, the sulfide groups of methionine (**38**) and biotin (**39, 40**) were unaffected under standard conditions with IPrCuCl catalyst. However, *S*-imidation of methionine (**38**) can proceed in high yield under condition **[C]** in which CuOAc was used as a catalyst[58].

In terms of the N partners, dioxazolones derived from both alkyl and aryl carboxylic acids can work well. Functional groups such as sulfoxide (**10**), NPhth (**12**), alkenyl (**9, 26**), alkynyl (**27**), azido (**28**), Fmoc (**17**), Cbz (**44**), unprotected indole (**43**), and OH (**41, 45**) were

## Table 2 | Metal-catalyzed *S*-amidation of primary thiol 4 with 2

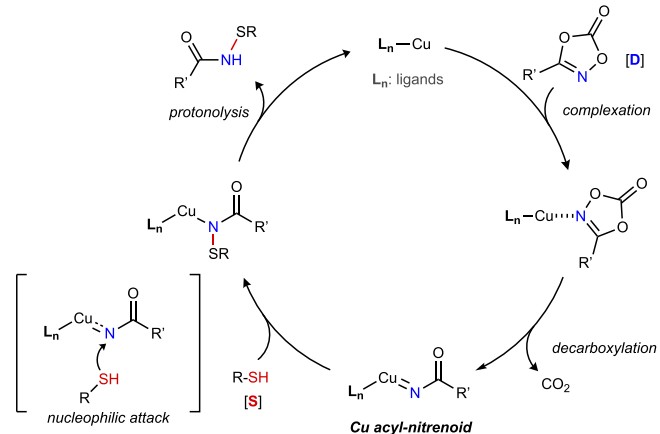

| Entry | Conditions | Yield (%)[a] 5a[b]/5b/5c |
|---|---|---|
| 1 | **4** (2 equiv), **2** (1 equiv), IPrCuCl (5 mol%) | 31/ND/69 |
| 2 | **4** (1 equiv), **2** (2 equiv), IPrCuCl (5 mol%) | 57/ND/42 |
| 3 | **4** (1 equiv), **2** (2 equiv), IPrCuCl (5 mol%), AgSbF$_6$ (5 mol%) | 61/ND/39 |
| 4 | **4** (1 equiv), **2** (2 equiv), IPrCuCl (5 mol%), AgBF$_4$ (5 mol%) | 65/ND/35 |
| 5 | **4** (1 equiv), **2** (2 equiv), IPrCuCl (5 mol%), NaBAr$^F_4$ (5 mol%) | 48/ND/52 |
| 6 | **4** (1 equiv), **2** (2 equiv), IPrCuCl (5 mol%), Ag$_2$CO$_3$ (5 mol%) | 69/ND/28 |
| 7 | **4** (1 equiv), **2** (2 equiv), IPrCuCl (5 mol%), Ag$_2$CO$_3$ (10 mol%) | 74(72[c])[b]/ND/22 |
| 8 | **4** (1 equiv), **2** (2 equiv), IPrCuCl (5 mol%), Ag$_2$CO$_3$ (20 mol%) | 53/ND/17 |
| 9 | **4** (1 equiv), **2** (2 equiv), Ag$_2$CO$_3$ (10 mol%) | ND/ND/ND |
| 10 | **4** (1 equiv), **2** (2 equiv), IPrAgCl (5 mol%) | ND/ND/ND |

*ND* not detected.

[a]All screening reactions were carried at a 0.2 mmol scale; yields are based on ¹H NMR analysis of crude reaction mixture.

[b]>99.9% ee for **5a**.

[c]Isolated yields at a 15.0 mmol scale.

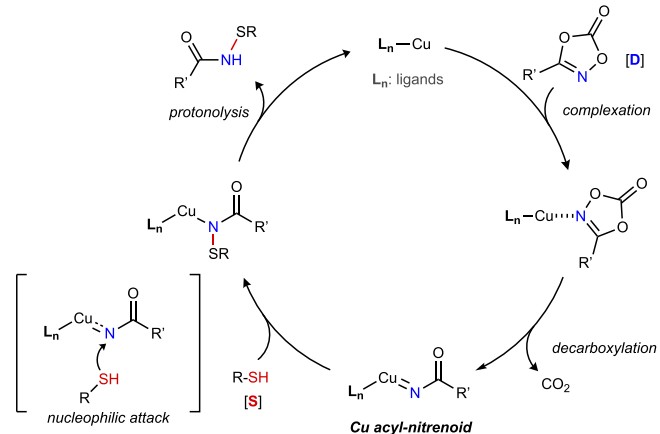

**Fig. 2 | Proposed mechanism of Cu-catalyzed *S*-amidation of thiols with diox-azolones.** The catalytic cycle features a sequence of decarboxylation of dioxazolone at Cu center, nucleophilic attack of thiol onto Cu-nitrenoid intermediate, and protonolysis.

well tolerated. As exemplified by **33**, the reaction of dioxazolone derived from α-amino acid with **4** gave the desired product **33** in 41% yield along with 49% of thiocarbamate side product **33'** formed by the addition of thiol to the rearranged isocyanate byproduct of dioxazolone (see Supplementary Fig. 12 for details). In comparison, reactions of dioxazolones bearing a protected amino group at more remote position from the α carbon of carboxylic acid can proceed in higher yield (**34**, **35**). As shown by **42–44**, cysteine-containing short peptides worked well. A cysteine carrying an amide linked fluorophore reacted with drug Lyrica-derived dioxazolones to give **41** in 82% yield. Mercapto-β-cyclodextrin (β-CD-SH) containing a number of free OH groups can be efficiently amidated to give **45** in 57% HPLC isolated yield.

### Utility of *N*-acetyl sulfenamides as *S*-sulfenylating reagents of thiols

The formation of disulfide side products during the synthesis of sulfenamides prompted us to explore their utility as a general *S*-sulfenylating reagent of thiols for the synthesis of unsymmetrical disulfides. The use of sulfenamides for disulfide synthesis has only been reported sporadically in the literature. Notably, Harpp showed that alkyl thiol phthalimide (alkyl-S-NPhth) prepared by the substitution of sulfenyl halide with phthalimide anion can form an unsymmetrical disulfide with cysteine in refluxed ethanol[78]. More recently, Shimizu showed *N*-trifluoroacetyl arenesulfenamide (Ar-S-NHCOCF$_3$) can react with thiols in organic solvents at rt to give disulfides bearing at least one aryl group[79]. However, applications of these methods have been limited due to the moderate reactivity and/or inconvenient access to these reagents.

We were pleased to find that *N*-acetyl sulfenamides can react with a range of primary and secondary thiols under mild conditions to give the corresponding unsymmetrical disulfides in good to excellent yield (Fig. 4). While most *N*-acyl sulfenamides can react, *N*-acetyl sulfenamides offered the optimal balance of reactivity, stability, and accessibility. The use of 1.5 equiv of *N*-acetyl sulfenamides was sufficient to achieve high conversion of thiols in most reactions. Simple acetamide AcNH$_2$ was generated as the only byproduct. The *S*-sulfenylation reactions with unhindered sulfenamides can proceed well in polar organic solvents like MeOH, MeCN, and THF at 35 °C (e.g., conditions **[D]**, **[E]**). These reactions also worked well in PBS buffer at pH 7.3 at lower concentration (0.01 M) and slightly elevated reaction temperatures (50 °C), which can help enhance the solubility of reactants in aqueous medium (condition **[F]**). Notably, aqueous media can promote reactions of more hindered sulfenamides. For example, compound **48** was obtained in excellent yields in PBS at 50 °C (condition **[F]**) or 1:1 mixed solvents of THF and PBS at 35 °C (condition **[G]**), whereas moderate yields of **48** were obtained in MeOH or THF (conditions **[D]**, **[E]**). As exemplified by compounds **47** and **51**, sulfenamide and disulfide formation can be carried out in a one-pot fashion without

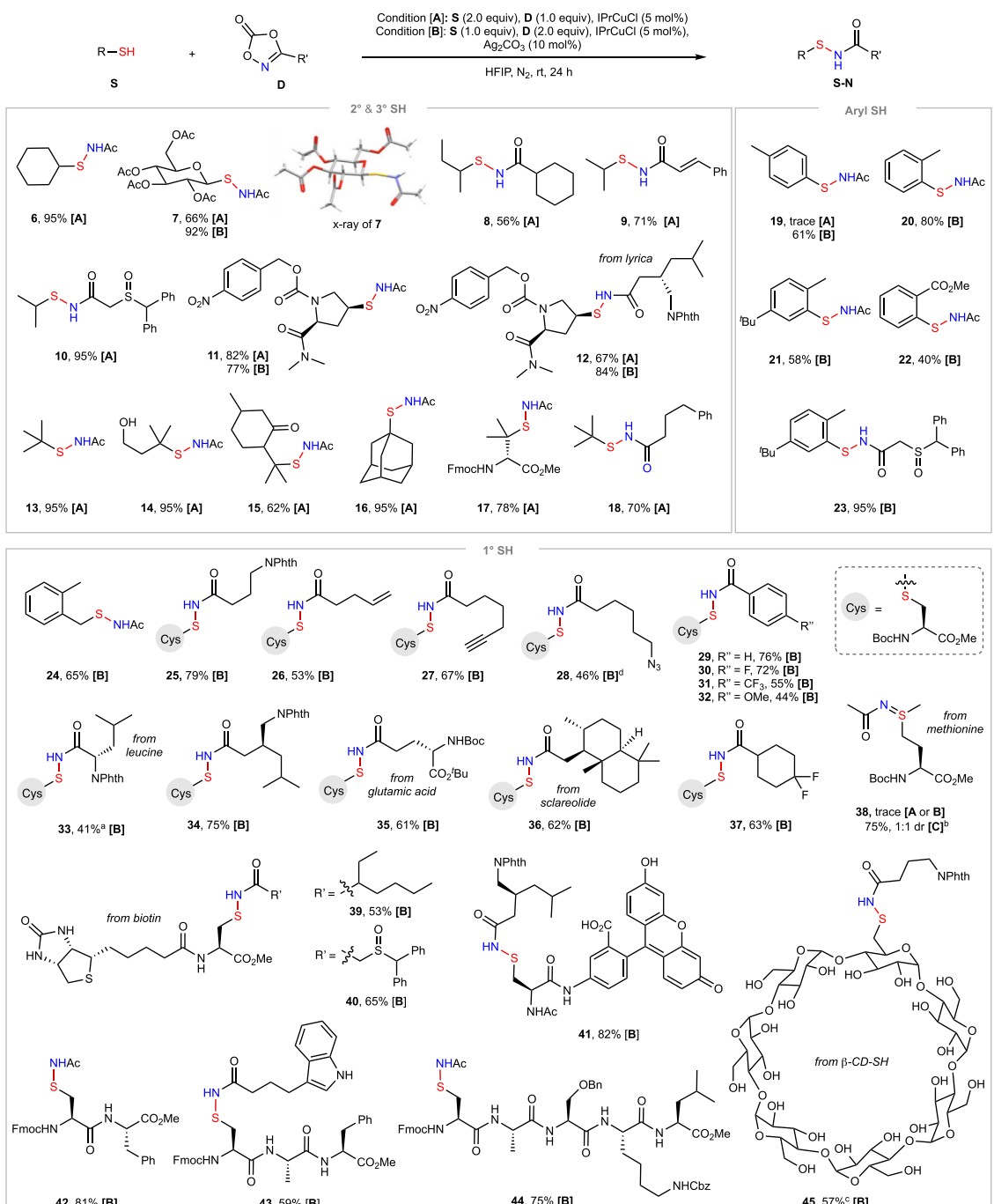

**Fig. 3 | Substrate scope of copper-catalyzed *S*-amidation of thiols with dioxazolones.** Isolated yield by silica gel column chromatography at 0.1–0.4 mmol scale. [a]Thiocarbamate side product **33'** was obtained in 49% yield. [b]Condition **[C]**: **S** (2

equiv), **D** (1 equiv), CuOAc (10 mol%), *n*-hexane (0.05 M), rt, 24 h. [c]HPLC isolated yield. [d]Disulfide **5c** was formed as the major side product.

purifying *N*-acetyl sulfenamide intermediates. In practice, HFIP solvent used in the first step needs to be swapped by evaporation under reduced pressure (see Supplementary Fig. 16 for details).

Versus other sulfenylating reagents such as sulfenyl chlorides, our *N*-acetyl sulfenamides showed excellent stability and compatibility with other nucleophiles. For example, the four acetate groups of *S*-glycosyl sulfenamide **7** can be cleanly removed by the treatment of K₂CO₃ in MeOH to give **7'**, which then reacted with a Cys-containing pentapeptide to give **57** carrying disulfide-linked free glucose in 80% LC-estimated yield (33% isolated yield by HPLC) under condition **[F]**. Amine, carboxylic acid, carboxamide, and indole side chain on

peptides were well tolerated (**50, 57**). A variety of functional moieties such as small molecule drugs (**49, 51**), biotin (**55**) and carbohydrates (**47, 54**) can be crosslinked. Short peptides can also be joined by disulfide bonds in high efficiency (**53, 55, 56**).

Synthesis of sterically hindered unsymmetrical disulfides bearing tertiary *S*-alkyl substituents remains challenging. The existing methods for this type of compound mainly rely on the disulfuration reactions of carbon partners with special disulfide-containing reagents. For example, Jiang recently reported a polar substitution reaction of *N*-dithiophthalimides (PhthN-SS-*tert*-alkyl) with 1,3-diketones nucleophiles[31]. Pratt showed that radical substitution of tetrasulfides

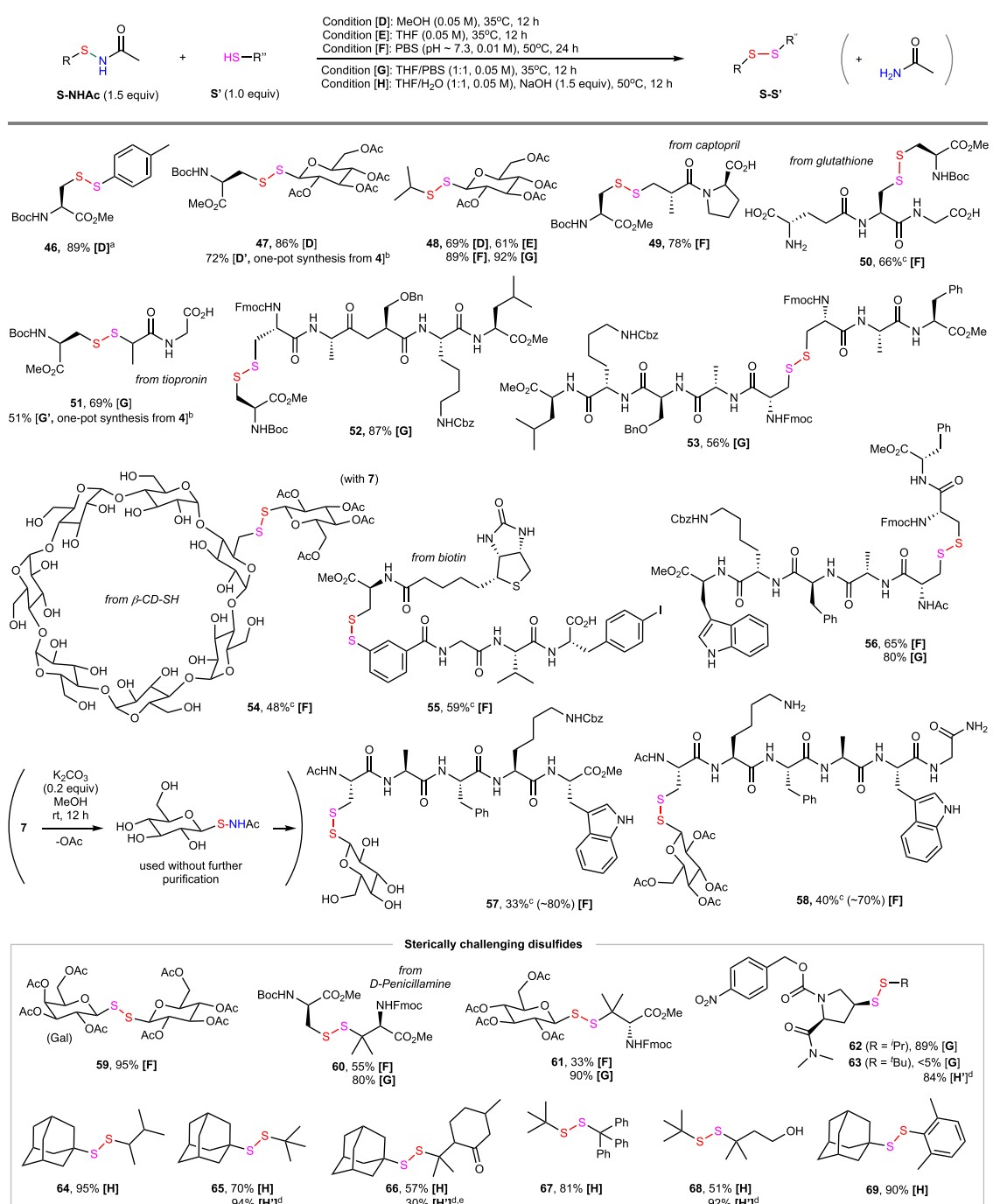

**Fig. 4 | The synthesis of unsymmetrical disulfides.** Isolated yield by silica gel column chromatography at 0.1 or 0.2 mmol scale unless specified otherwise. LC-estimated yield for selected products were shown in parenthesis. [a]>99.9% ee. [b]Disulfides were synthesized from the corresponding RSH in a one-pot fashion without purifying sulfenamide intermediates. HFIP solvent was swapped by evaporation under reduced pressure. In the modified conditions [D'] and [G'], 1.5 equiv of R"SH was used. [c]HPLC isolated yield. [d]Condition [H'] is same as condition [H] except a 1:2 ratio of **S-NHAc** and **S'**. [e]N-acetyl sulfenamide of *para*-mentha-8-thiol-3-one is relatively unstable. PBS phosphate buffered saline.

(R-SSSS-R) and trisulide-1,1-dioxides (PhSO₂-SS-alkyl) with tertiary alkyl radicals can proceed in high yield under thermal or photocatalytic conditions[32,33]. In comparison, convergent methods for disulfide synthesis by joining two bulky thiol partners are much under-developed. We were pleased to find that *N*-acetyl sulfenamides bearing bulky *S*-substituents can react with other bulky thiols to form the corresponding disulfides in moderate to excellent yield under mild conditions. For example, compound **61** bearing one secondary and one tertiary *S*-alkyl substituents was obtained in 90% yield under

neutral condition [G]. Notably, β-thiol carbonyl compounds such as D-penicillamine for products **60** and **61** can undergo β-elimination to form alkene side products when the reactions were conducted under basic conditions. As exemplified by **65**, unsymmetrical disulfides bearing two tertiary *S*-alkyl groups can be formed in good yield in mixed solvents of THF and water with the addition of 1.5 equiv of NaOH at 50 °C (condition [H]). Small amounts of decomposition products of *N*-acetyl sulfenamides were observed under these reaction conditions. The use of excess thiols (2 equiv) can further increase the reaction

yield (condition **[H']**, see **65**, **68**). Product **69** features one tertiary alkyl and one bulky aryl substituent and was obtained in 90% yield under condition **[H]**). Control experiments showed that unhindered unsymmetrical disulfides such as **47** can be obtained in moderate yield and chemoselectivity via direct coupling of two thiols under selected oxidative conditions[80–83]. In comparison, the yield and selectivity for sterically demanding unsymmetrical disulfides such as **65** significantly diminished under the same treatments (see Supplementary Fig. 17 for details).

In summary, we developed a nitrene-mediated *S*-amidation reaction of thiols under copper catalysis. These copper-catalyzed reactions offer an efficient, convenient, and broadly applicable method for the preparation of *N*-acyl sulfenamides from readily accessible precursors. Moreover, the resulting *N*-acetyl sulfenamides provide a class of highly effective *S*-sulfenylation reagents for the synthesis of unsymmetrical disulfides under mild conditions. The *S*-sulfenation protocol enables facile access to many sterically demanding disulfides that are difficult to prepare by other means.

## Methods

### Typical procedure for the Cu-catalyzed *S*-amidation of thiols with dioxazolones under standard conditions

To a solution of Boc-protected cysteine methyl ester **4** (3.53 g, 15 mmol, 1.0 equiv), IPrCuCl (366 mg, 0.75 mmol, 5 mol%) and $Ag_2CO_3$ (414 mg, 1.5 mmol, 10 mol%) in HFIP (37.5 mL, 0.4 M), 3-methyldioxazolone **2** (30 mmol, 3.03 g, 2.0 equiv) was added. The reaction mixture was stirred under $N_2$ atmosphere for 24 h at rt. The reaction mixture was then concentrated under reduced pressure, redissolved in DCM (50 mL), and washed with $H_2O$ (50 mL). The organic layer was dried with anhydrous $Na_2SO_4$ and concentrated *in vacuo*. The resulting residue was purified by silica gel flash chromatography using hexanes/EtOAc (10/1 to 3:1, v/v) eluent to give the desired product **5a** as a white solid (3.15 g, 72% yield).

### Typical procedure for the disulfide synthesis using *N*-acetyl sulfenamides as the *S*-sulfenylating reagents of thiols

To the solution of sulfenamide **13** (29.4 mg, 0.2 mmol, 1.0 equiv) and 3-mercapto-3-methylbutan-1-ol (48.1 mg, 0.4 mmol, 2.0 equiv) in THF (2.0 mL), aqueous solution of NaOH (12.0 mg, 0.3 mmol, 1.5 equiv in 2.0 mL of water) were added. The reaction mixture was stirred at 50 °C under air for 12 h. The reaction mixture was concentrated *in vacuo*. The resulting residue was purified by silica gel flash chromatography using hexanes/EtOAc eluent (20/1 to 5:1, v/v) to give the desired product **68** as a colorless oil (38.3 mg, 92% yield).

## Data availability

The X-ray crystallographic data for compound **7** have been deposited in the Cambridge Crystallographic Data Centre with a number of CCDC: 2151305, and can be obtained free of charge from the CCDC via www.ccdc.cam.ac.uk/getstructures. Detailed synthetic procedures, additional control experiments, NMR spectra, and LC-MS spectra are available within the main article and its Supplementary information.

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

## Acknowledgements

This work was partially supported by the National Natural Science Foundation of China (21725204, 21901127), Frontiers Science Center for New Organic Matter (63181206), the China Postdoctoral Science Foundation (2018M640225, 2019T120179), and Haihe Laboratory of Sustainable Chemical Transformations.

## Author contributions

Z.B. made the initial discovery of the project and finished most of the experiments. S.Z. participated in the discussion of the subject. Y.H. helped with the expansion of the substrate scope. P.Y. and X.C. provided part of the peptide substrates and participated in the corresponding discussion. G.H. supervised part of the synthetic studies. H.W. gave the initial idea and supervised part of the synthetic studies. G.C. supervised the entire project and prepared most of the manuscript.

## Competing interests

The authors declare no competing interests.

## Additional information

**Correspondence and requests** for materials should be addressed to Hao Wang or Gong Chen.

