## [Peer Review File · Nature Communications]

N-acyl sulfenamides via copper-catalyzed S-amidation of thiols with dioxazolones for synthesis of unsymmetrical disulfidesREVIEWER COMMENTS

Reviewer #1 (Remarks to the Author):

Chen and coworkers have developed a new method for synthesis of N-acyl sulfenamides via copper-catalyzed nitrene-mediated S-amidation reaction of thiols with dioxazolones. And the resulting N-acetyl sulfenamides are highly effective sulfenylation reagents for the synthesis of unsymmetrical disulfides under mild conditions. Compared with the previous method, this method prepared a new N-acyl sulfenamides reagent, through this reagent, various disulfide compounds can be constructed gently and quickly, and a variety of peptide disulfide compounds can be synthesized, as well as various high sterically hindered disulfide compounds that are difficult to obtain before can be synthesized efficiently. The following issues should be addressed before the acceptance of the manuscript for publishing.

- 1) The author mentioned primary thiols have high nucleophilicity and could more easily react with sulfenamides to form a disulfide side product than secondary and tertiary thiols. When using primary thiols as substrate to prepare N-acyl sulfenamides reagent, the author added catalytic amount Ag₂CO₃. Can the author explain the role of silver catalyst in this reaction?
- 2) The author's ultimate goal is to obtain disulfide compounds. Does the author consider that after obtaining N-acyl sulfenamides, which can directly input it into the next reaction without further purification, so that different disulfide compounds can be obtained more efficiently, which can better illustrate the advantages of this reagent and method.
- 3) Product 38, S-imidation of methionine (38) can proceed in high yield under conditions which CuOAc was used as a catalyst. Can the author explain why such a product is obtained under such conditions?

Reviewer #2 (Remarks to the Author):

The manuscript entitled "Copper-catalyzed Amidation of Thiols with Dioxazolones: New Sulfenamide Reagents for Synthesis of Unsymmetrical Disulfides" (NCOMMS-22-22200) is well written and after minor revision can be published in a journal such as Nature Communications.

The strongest part of manuscript is based on the development of conditions, reagents and catalyst for direct preparation of N-acyl sulfenamides under mild conditions and wide scope of tolerated functionalities. The acceptable and very good yield is observed for primary secondary, tertiary, aromatic thiols and cysteine derivatives as well. However, in the case of cysteine derivatives for example 25, 26, or 27 the optical purity was not determined. Moreover other chiral compounds were not examined from that point of view as well. In my opinion the retention of configuration for selected example should be presented. The selection of acyl groups is almost perfect, the terminal azide group at acyl functionality would complete this selection. The presence of double and triple C-C bond, biotin, CD and peptides provided excellent platform for biological applications.

N-acyl sulfenamides were subsequently used for unsymmetrical disulfides preparation with excellent results. However, retention of configuration for selected chiral disulfide should be demonstrated (compound 45 or 50 are convenient).

From my point of view the developed method for preparation of sulfenamides and unsymmetrical disulfides is the best so far available in the literature.

The quality of the data is very good as it supposed to be. The spectra are correct and interpreted carefully in sufficient detail. All conclusion are justified and supported with strong experimental results. The potential significance of the results is the preparation of unsymmetrical disulfides by modern and effective method what is devoted to the advancement of the science of synthetic chemistry, covering fields of medicinal, biological, and surface chemistry.

Editorial remarks:

1. The most comprehensive reviews of S-S bond synthesis are included below and should be considered by authors (one of them is already included in ref 8 (g)):

- (a) Shcherbakova, I.; Pozharskii, A. F., In *Comprehensive Organic Functional Group Transformations II*, Katritzky, A. R.; Taylor, R.; Ramsden, Ch., Eds.; Pergamon: Oxford, 2004; Vol. 2, pp 210–233;
- (b) Bulman Page, P. C.; Wilkes R. D.; Reynolds, D., In *Comprehensive Organic Functional Group Transformations*, Katritzky, A. R.; Meth-Cohn, O.; Rees, C. W., Eds.; Pergamon: Oxford, 1995; Vol. 2, pp 177–187.
- (c) Sato, R.; Kimura, T., In *Science of Synthesis*, Kambe, N.; Drabowicz, J.; Molander, G. A., Eds.; Thieme: Stuttgart – New York, 2007; Vol. 39, pp 573-588; (available also on line)
- (d) Witt, D.; Klajn, R.; Barski, P.; Grzybowski, B.A. *Curr. Org. Chem.*, 2004, 8, 1763.
- (e) Mandal, B. and Basu, B., *RSC Adv.*, 2014, 4, 13854-13881.
- (f) Witt, D. *Synthesis* 2008, 2491-2509.
- (g) Musiejuk, M. and Witt, D. *Org. Prep. Proced. Int.* 2015, 47, 95-131
- The synthesis of sulfenamides described in PHOSPHORUS, SULFUR, AND SILICON 2016, VOL. 191, NO. 2, 305-310 can be also considered by authors.
2. The italic letter S should be used for S-alkyl or S-imidation in manuscript.

In the light of the above comments to Author, I do find enough novelty and urgency to publish this work in *Nature Communications* after minor revision.

Congratulations to Authors, the manuscript is one of the most interesting paper that I have read recently.

Reviewer #3 (Remarks to the Author):

Prof. Chen, Wang and coworkers developed a Copper-catalyzed amidation of thiols with dioxazolones for the synthesis of sulfenamide and applied the observed sulfenamide for the construction of unsymmetrical disulfides. It is a great challenge to synthesis of unsymmetrical disulfides—especially those with significant steric hindrance around the α carbons. This study provided a new method for the synthesis of N-acyl sulfenamides under mild conditions. Moreover, the resulting N-acetyl sulfenamides works as an ideal powerful S-sulfenylating reagents for the synthesis of challenging unsymmetrical disulfides. The paper is well prepared with broad substrate scope. I recommend it to be published in NC after minor revision.

1. It is better to provide the reaction mechanism for the Copper-catalyzed amidation of thiols with dioxazolones.
2. Why 38 was obtained as S-imidation of methionine instead of S-amidation?
3. For example, compounds 46 and 64, they were prepared from the corresponding thiols via two steps. How about the direct oxidative coupling of thiols? The results of the comparison need to be provided.

Dear reviewers,

Thank you very much for the supportive and constructive comments to our work. Your suggestion has greatly helped us to improve the quality of this manuscript. We have conducted new experiments and made proper revisions in both main text and supporting information to address your concerns. Please see our point-by-point response below.

Reviewer #1

Chen and coworkers have developed a new method for synthesis of N-acyl sulfenamides via copper-catalyzed nitrene-mediated S-amidation reaction of thiols with dioxazolones. And the resulting N-acetyl sulfenamides are highly effective sulfenylation reagents for the synthesis of unsymmetrical disulfides under mild conditions. Compared with the previous method, this method prepared a new N-acyl sulfenamides reagent, through this reagent, various disulfide compounds can be constructed gently and quickly, and a variety of peptide disulfide compounds can be synthesized, as well as various high sterically hindered disulfide compounds that are difficult to obtain before can be synthesized efficiently. The following issues should be addressed before the acceptance of the manuscript for publishing.

Q1. 1) *The author mentioned primary thiols have high nucleophilicity and could more easily react with sulfenamides to form a disulfide side product than secondary and tertiary thiols. When using primary thiols as substrate to prepare N-acyl sulfenamides reagent, the author added catalytic amount Ag_2CO_3 . Can the author explain the role of silver catalyst in this reaction?*

A1. The role of Ag_2CO_3 additive is not very clear at the moment. In general, it can improve the yield of desired products by 10-20%. We suspect that Ag_2CO_3 may slightly lower the concentration of free thiol reactant by reversible complexation, thus reducing the formation of undesired disulfide side product. The above discussion has been added to the main text (see page 6, line 125-127).

Q2. 2) *The author's ultimate goal is to obtain disulfide compounds. Does the author consider that after obtaining N-acyl sulfenamides, which can directly input it into the next reaction without further purification, so that different disulfide compounds can be obtained more efficiently, which can better illustrate the advantages of this reagent and method.*

A2. It is indeed possible to carry out the two steps in one pot, which gave slightly improved overall yield of disulfide products. Since different solvents were used in the two steps, a simple solvent swap operation is needed. In practice, HFIP solvent can be readily removed under reduced pressure after the sulfenamide synthesis step, the other thiol reactant and solvent were

then added for the disulfide formation. Representative results for one-pot synthesis of compounds **47** and **51** have been added to the main text, Figure 4 and SI (page 9, line 180-183).

Q3. 3) *Product 38, S-imidation of methionine (38) can proceed in high yield under conditions which CuOAc was used as a catalyst. Can the author explain why such a product is obtained under such conditions?*

A3. Our initial experiments showed that the S-amination of thiols with dioxazolones can proceed with high reactivity under the catalysis of a variety of metal complexes (see Table 1). Besides reacting with thiols to generate disulfides, the desired sulfenamide products can undergo further S-amination to give sulfinamidines, leading to the formation of a complex mixture. This work showed the reactivity of nitrene-mediated S-amination of thiols can be tamed with the use of NHC ligands to selectively form sulfenamide. The NHC ligand significantly reduces the reactivity of sulfide of methionine with the Cu-nitrenoid intermediate. In comparison, the catalysis of simple CuOAc in hexane without the mediation of NHC ligand can enable the S-amination of sulfides to give the corresponding sulfilimine products. A related discussion has been included in the main text: The inherent reactivity of the NHC-bound Cu-nitrenoid intermediate might be critical to preventing further S-imidation of the N-acyl sulfenamides (page 7, line 134-135).

Reviewer #2

The manuscript entitled “Copper-catalyzed Amidation of Thiols with Dioxazolones: New Sulfenamide Reagents for Synthesis of Unsymmetrical Disulfides” (NCOMMS-22-22200) is well written and after minor revision can be published in a journal such as *Nature Communications*. The strongest part of manuscript is based on the development of conditions, reagents and catalyst for direct preparation of N-acyl sulfenamides under mild conditions and wide scope of tolerated functionalities. The acceptable and very good yield is observed for primary secondary, tertiary, aromatic thiols and cysteine derivatives as well.

Q1. However, in the case of cysteine derivatives for example 25, 26, or 27 the optical purity was not determined. Moreover other chiral compounds were not examined from that point of view as well. In my opinion the retention of configuration for selected example should be presented.

A1. Results of selected cysteine-derived products showed complete retention of chiral integrity in sulfenamides and disulfides (>99.9% ee of **5a** and **46** based on the comparison with racemic mixtures by chiral HPLC analysis). These results have been added to the main text, Table 2, and Figure 4 (see page 6, line 123; page 20, line 468).

Retention of configuration of cysteine derivatives in the S-N and S-S coupling

Q2. *The selection of acyl groups is almost perfect, the terminal azide group at acyl functionality would complete this selection. The presence of double and triple C-C bond, biotin, CD and peptides provided excellent platform for biological applications.*

A2. Compound **28** carrying an azido group was added to the revised Figure 3. It was isolated in 46% yield along with the formation of disulfide as the major side product.

Q3. *N-acyl sulfenamides were subsequently used for unsymmetrical disulfides preparation with excellent results. However, retention of configuration for selected chiral disulfide should be demonstrated (compound 45 or 50 are convenient).*

A3. This question has been addressed in the answer to Q1 (see above). The stereochemistry of cysteine-derived products was unaffected.

From my point of view the developed method for preparation of sulfenamides and unsymmetrical disulfides is the best so far available in the literature.

The quality of the data is very good as it supposed to be. The spectra are correct and interpreted carefully in sufficient detail. All conclusion are justified and supported with strong experimental results.

The potential significance of the results is the preparation of unsymmetrical disulfides by modern and effective method what is devoted to the advancement of the science of synthetic chemistry, covering fields of medicinal, biological, and surface chemistry.

Editorial remarks:

Q4. *1. The most comprehensive reviews of S-S bond synthesis are included below and should be considered by authors (one of them is already included in ref 8 (g)):*

(a) Shcherbakova, I.; Pozharskii, A. F., In Comprehensive Organic Functional Group Transformations II, Katritzky, A. R.; Taylor, R.; Ramsden, Ch., Eds.; Pergamon: Oxford, 2004; Vol. 2, pp 210–233;

(b) Bulman Page, P. C.; Wilkes R. D.; Reynolds, D., In Comprehensive Organic Functional Group Transformations, Katritzky, A. R.; Meth-Cohn, O.; Rees, C. W., Eds.; Pergamon: Oxford, 1995; Vol. 2, pp 177–187.

(c) Sato, R.; Kimura, T., In Science of Synthesis, Kambe, N.; Drabowicz, J.; Molander, G. A., Eds.; Thieme: Stuttgart – New York, 2007; Vol. 39, pp 573-588; (available also on line)

(d) Witt, D.; Klajn, R.; Barski, P.; Grzybowski, B.A. Curr. Org. Chem., 2004, 8, 1763.

(e) Mandal, B. and Basu, B., RSC Adv., 2014, 4, 13854-13881.

(f) Witt, D. Synthesis 2008, 2491-2509.

(g) Musiejuk, M. and Witt, D. Org. Prep. Proced. Int. 2015, 47, 95-131

The synthesis of sulfenamides described in PHOSPHORUS, SULFUR, AND SILICON 2016, VOL. 191, NO. 2, 305-310 can be also considered by authors.

A4. These references have been added to the main text.

Q5. *2. The italic letter S should be used for S-alkyl or S-imidation in manuscript.*

A5. This issue has been addressed in the revised main text.

In the light of the above comments to Author, I do find enough novelty and urgency to publish this work in Nature Communications after minor revision.

Congratulations to Authors, the manuscript is one of the most interesting paper that I have read recently.

Reviewer #3:

Prof. Chen, Wang and coworkers developed a Copper-catalyzed amidation of thiols with dioxazolones for the synthesis of sulfenamide and applied the observed sulfenamide for the construction of unsymmetrical disulfides. It is a great challenge to synthesis of unsymmetrical disulfides—especially those with significant steric hindrance around the α carbons. This study provided a new method for the synthesis of N-acyl sulfenamides under mild conditions. Moreover, the resulting N-acetyl sulfenamides works as an ideal powerful S-sulfonylating reagents for the synthesis of challenging unsymmetrical disulfides. The paper is well prepared with broad substrate scope. I recommend it to be published in NC after minor revision.

Q1. 1. It is better to provide the reaction mechanism for the Copper-catalyzed amidation of thiols with dioxazolones.

A1. We have included more detailed mechanistic discussions on the Cu-catalyzed S-amidation reaction (page 6, line 131-134). Figure 2 describing our proposed mechanistic pathway was also added.

Q2. 2. Why 38 was obtained as S-imidation of methionine instead of S-amidation?

A2: Please see our answer to Q3 of reviewer 1.

Q3. 3. For example, compounds 47 and 65, they were prepared from the corresponding thiols via two steps. How about the direct oxidative coupling of thiols? The results of the comparison need to be provided.

A3. Two different thiols can be directly coupled to form disulfides under oxidative conditions. However, controlling hetero-selectivity is usually problematic. Syntheses of representative compounds **45** and **65** for unhindered and hindered disulfide respectively were tested under four known oxidative coupling conditions. **45** can be obtained in moderate yields and selectivity. In comparison, the yield and selectivity for sterically hindered **65** significantly dropped. Please see below for the summary of the results, which have been added to the revised SI. The following discussions have been added to main text: Control experiments showed that unhindered unsymmetrical disulfides such as **45** can be obtained in moderate yield and chemoselectivity via direct coupling of two thiols under selected oxidative conditions.⁸⁰⁻⁸³ In comparison, the yield and selectivity for sterically demanding unsymmetrical disulfides such as **65** significantly diminished under the same treatments (see supplementary Fig. 17 for details).

Comparisons of oxidative coupling of thiols with our methods:

trace [I]
 30% [J]
 57% [K]
 22% [L]

Oxidative coupling **89% [D]** **Our method**

7% [I]
 trace [J]
 trace [K]
 trace [L]

Oxidative coupling **65** **94% [H']** **Our method**

Thanks again for your helpful suggestions. We will be happy to make additional revisions to satisfy your concerns if needed.

Sincerely,

Gong Chen, Ph.D.
 Hao Wang, Ph.D.
 Nankai University

REVIEWERS' COMMENTS

Reviewer #1 (Remarks to the Author):

The authors have answered all the questions raised by the three reviewers appropriately. Thus this manuscript could be accepted for publication in current form.

Reviewer #3 (Remarks to the Author):

The authors revised the manuscript carefully, which meet the requirement of NC. I recommend it to be published in NC.